# The Importance of Solution Studies for the Structural Characterization of the Enterovirus 5’ Cloverleaf

**DOI:** 10.3390/v17081127

**Published:** 2025-08-16

**Authors:** Morgan G. Daniels, Meagan E. Werner, Xiaobing Zuo, Steven M. Pascal

**Affiliations:** 1Department of Chemistry and Biochemistry, Old Dominion University, Norfolk, VA 23529, USA; mdani006@odu.edu (M.G.D.); mwern002@odu.edu (M.E.W.); 2X-Ray Science Division, Argonne National Laboratory, Lemont, IL 60439, USA; zuox@anl.gov

**Keywords:** RNA, enterovirus, picornavirus, cloverleaf, replication

## Abstract

Enteroviruses initiate genomic replication via a highly conserved mechanism that is controlled by an RNA platform, also known as the 5’ cloverleaf (5’CL). Here, we present a biophysical analysis of the 5’CL conformation of three enterovirus serotypes under various ionic conditions, utilizing CD spectroscopy, size-exclusion chromatography, and small-angle X-ray scattering. In general, a tendency toward a smaller monomeric hydrodynamic radius in the presence of salts was observed, but the exact structural signature of each 5’CL varied depending upon the serotype. Rhinovirus B14 (RVB14) exhibited at least two monomeric conformations and a low propensity for dimerization, while poliovirus 1 (PV1) showed a high propensity for dimerization, which was enhanced by the presence of salts. Enterovirus D70 was observed to be somewhat intermediate, with primarily a monomeric structure, but possessing some potential for dimerization. The equilibrium between the two monomeric and the dimeric conformations is also discussed. These results indicate that the 5’CL conformation may be more complex than the current literature suggests, thus underscoring the need for a combined crystal and solution approach for the accurate representation of the 5’CL conformation, and the conformation of other RNA structural elements, under native conditions.

## 1. Introduction

The enterovirus genus, a subset of the picornavirus family, contains over 300 virus serotypes [1,2]. Enteroviruses are responsible for various human illnesses, including polio, viral myocarditis, acute hemorrhagic conjunctivitis, and the common cold [3,4,5]. Despite the high prevalence of enterovirus infections, only four licensed vaccines are available, and no antivirals currently exist [6].

The enterovirus RNA genome consists of a small (~7.5 kb) positive-sense RNA strand with three regions: a main open reading frame (ORF) flanked by two non-translated regions (NTRs), as illustrated in Figure 1 [1,7]. The ORF encodes a ~250 kDa polyprotein that, after post-translational processing, produces up to 11 mature proteins. However, various partially cleaved polyproteins play essential roles in the viral life cycle [8,9]. For example, the 3CD polyprotein plays a critical role in replication; it encompasses both the primary viral protease (3C^pro^) and the viral replicase (3D^pol^). The two NTRs form specific structural elements that play key roles in the viral life cycle. Comprising 37–99 nucleotides, the 3’-NTR contains two stem loops (domain X and Y) and a poly-A tail [10,11].

The 5’-NTR is much larger: it contains ~750 nucleotides that fold to form the 5’ cloverleaf (5’CL) and the internal ribosome entry site (IRES), which are essential for the initiation of replication and translation, respectively [12]. The 5’CL RNA is formed by the first ~100 nucleotides of the enterovirus RNA genome. The cloverleaf (CL) designation was coined from secondary structure predictions (see Figure 2) that showed three stem-loops (SLB, SLC, and SLD) connected through a junction to one stem (SA). This platform serves as a locus for viral and host proteins to gather and initiate virus replication [13,14,15]. For example, the host poly-C binding protein (PCBP2) interacts with a C-rich region located in the loop of SLB [16,17]. Once bound, PCBP2 can interact with the host poly-A binding protein (PABP), which binds to the poly-A tail from the 3’-NTR, thus effectively circularizing the viral genome [18]. The binding of two additional viral proteins (VPg and 3CD) to the 5’CL is also essential to viral replication [19,20,21,22].

Here, we explored the effect of cations on the solution conformation of the 5’CL from three enterovirus serotypes. Circular dichroism (CD) was used as an initial screen for structural changes. The conformational species present in solution were examined via size-exclusion chromatography (SEC) separation, which is based on both size and shape. Finally, SEC-SAXS was used to characterize the dominant conformations found in solution and to detect dimerization. The results suggest that a mixture of conformations, the composition of which is dependent upon both the ionic conditions and the serotype, can be present in solution. These results demonstrate the importance of including a solutions approach in the structural characterization of the enterovirus 5’CL RNA and other RNA structural elements.

## 2. Materials and Methods

### 2.1. Production of 5’CL via T7 In Vitro Transcription

Bacteriophage T7 RNA polymerase, made in-house, utilizes double-stranded DNA as a template for in vitro transcription. Plasmids purchased from SynbioTechnologies (Monmouth Junction, NJ, USA) included the T7 RNA promoter sequence; a cis-acting hammerhead ribozyme; the 5’CL sequence from RVB14, PV1, or EVD70; and a BsaI restriction site. E. coli (DH5a) cells were transformed with the plasmid and cultured to mass produce the DNA plasmid, which was then extracted, purified, and linearized via the BsaI cut site. In vitro transcription using this linearized DNA as a template utilized a transcription buffer (400 mM Tris-HCl, 128 mM MgCl_2_, 10 mM DTT, and 20 mM spermidine, with a pH of 7.9) and 5 M rNTPs (Jena Biosciences) to produce each 5’CL RNA sample. The large-scale transcript, 10 mL per reaction, was incubated overnight at 37 °C prior to purification via gel electrophoresis in a 10% urea-PAGE gel. The band of interest was excised from the gel, crushed through a syringe, and then soaked in crush and soak buffer (10 mM NaPi and 0.1 mM EDTA) overnight at 4 °C. The supernatant was concentrated and buffer exchanged via ethanol precipitation and a spin concentrator (cutoff = ~10 kDa). The purified RNA was pelleted via ethanol precipitation, and the resulting pellet was resuspended in one of four buffer conditions, as shown in Table 1, and brought to the needed concentration, as shown in Table 2, for the analysis technique.

### 2.2. Circular Dichroism

CD was performed using a J-815 CD spectrometer to fingerprint structural changes in the 5’CL samples due to the changing ionic environment. The CD samples were prepared to an RNA concentration of 0.0015 mM in 10 mM NaPi and 0.1 mM EDTA, and adjusted to varying concentrations of potassium chloride and magnesium chloride, as shown in Table 1. The samples were then passed through a 0.22 µm filter and loaded into a 1 mm quartz cuvette. CD data were acquired from 320 nm to 200 nm with a scan speed of 20 nm/min at 25 °C. The data were processed with buffer blank subtraction, smoothing using a means-movement function of 25 nm, and conversion to molar ellipticity.

### 2.3. Size-Exclusion Chromatography—Superdex 75

The SEC samples were concentrated to 0.5 mg·mL^−1^ in 10 mM NaPi and 0.1 mM EDTA, along with varying concentrations of potassium chloride and magnesium chloride, as shown in Table 1, and passed through a 0.22 µm filter. The samples were loaded into the capillary loop of the AKTA liquid chromatography system. A Superdex 75 10/300 GL (CV = 23.562 mL) was equilibrated at 0.5 mg·mL^−1^ using 3 CVs of the matching buffer. After loading the sample onto the column, isocratic elution was performed with 1 CV at a rate of 0.5 mg·mL^−1^, and the resulting output was collected in 5.00 mL fractions. The resulting chromatograms will be referred to as SEC75 for the remainder of this paper.

### 2.4. Size-Exclusion Chromatography—Small-Angle X-Ray Scattering (SEC-SAXS)

Samples designated for SEC-SAXS were concentrated to 5.5 mg·mL^−1^ with 140 K + 2 Mg ionic conditions, as seen in Table 1, and passed through a 0.22 µm filter.

SEC-SAXS was performed at beamline 12-ID-B of the Advanced Photon Source at the Argonne National Laboratory using an in-line FPLC setup (AKTA micro) equipped with a Superdex 200 Increase 10/300 GL size-exclusion column. The wavelength, λ, of X-ray radiation was set to 0.932 Å. Scattering data were collected using an Eiger2 9 M detector (DECTRIS LLC). The sample-to-detector distance was set to cover a momentum transfer (q) range of 0.004–0.85 Å^−1^, where q = 4π sinθ/λ, and 2θ is the scattering angle. Samples eluting from the SEC column flowed directly into a quartz capillary flow cell for SAXS measurements. The cylindrical quartz capillary had a diameter of 1.5 mm and a wall thickness of 10 μm. Two-dimensional SAXS images were recorded every 2 s with an exposure time of 0.5 s to reduce radiation damage. The 2D scattering images were converted to 1D SAXS (I(q) vs. q) curves through azimuthally averaging after a solid angle correction and then normalizing with the intensity of the transmitted X-ray beam flux, using the beamline software package matSAXS (https://12idb.xray.aps.anl.gov/Software_Processing.html. Accessed on 1 August 2025). All of the SEC peaks chosen for the SAXS analysis were fairly well resolved. More than twenty SAXS frames around the elution peak were averaged to obtain the sample SAXS profile. Background profiles were obtained by averaging fifty frames from the flat baseline region before or after the peaks. The final sample SAXS profiles were obtained by subtracting the background from the corresponding sample data.

## 3. Results

This section is divided by subheadings. It should provide a concise and precise description of the experimental results and their interpretation, as well as the experimental conclusions that can be drawn.

### 3.1. CD Spectroscopy Showed Variation with Ionic Conditions

CD spectroscopy (Figure 3) provided evidence of conformational differences in the four different experimental conditions of this study. In general, the global maximum near 265 nm tended to increase in intensity and shift left upon the addition of salts. Similarly, the global minimum near 210 mL tended to increase in intensity upon the addition of salts. Both of these changes indicate alterations in base stacking that would accompany conformational changes. While it is difficult to interpret in detail the structural changes giving rise to the observed spectral changes, it is clear that the ionic environment affects the 5’CL structure in general, and that those changes that are trackable via CD spectroscopy are largely shared by the three serotypes tested.

However, the three serotypes did display some distinctions. For instance, the RVB14 5’CL (Figure 3A) exhibited a negligible change in the intensity of the maximum unless K and Mg were combined. Also, both RVB14 and EVD70 (Figure 3C) appeared to be more strongly affected by Mg than by K. This may be due to the greater ability of the divalent Mg ion to concentrate positive charge. In contrast, PV1 (Figure 3B) showed a stronger response to K than to Mg. This is likely explainable by the higher concentration of K (140 mM) compared to Mg (5 mM): if equivalent ionic concentrations were used, the response of PV1 to Mg would likely be stronger than its response to K. Appendix A further explores these cation-induced shifts via CD difference spectra.

### 3.2. SEC Showed Multiple Conformational Species

Size-exclusion chromatography (SEC) was used to further examine the effect of the ionic environment on the 5’CL conformation. SEC is sensitive not only to size (molecular weight), but also to conformation (hydrodynamic radius) as well as self-association state (e.g., dimerization). Figure 4 shows the SEC chromatograms of each of the three 5’CL sequences under the conditions of Table 1. Consistent with the CD results, the SEC results varied significantly as the ionic environment varied. In addition, we observed distinct patterns of change for the three serotypes tested. Interestingly, we observed multiple SEC peaks under most conditions, indicating multiple conformational species. This suggests that the CD results are a weighted average of the multiple conformations present in solution. However, the SEC experiments were performed at approximately 10× the concentration of the CD experiments (see Table 1), and thus, dimerization, if present, may have been more prevalent under the SEC conditions.

### 3.3. Native and Denaturing Gel Electrophoresis to Augment the SEC Analysis

To further assay the monomer–dimer equilibrium, native gel electrophoresis was utilized. Note that filtration (see the Materials and Methods) was used immediately prior to all the experiments in this manuscript except for gel electrophoresis. Therefore, the largest species detected by native gel electrophoresis would have been mostly absent from the samples used for CD, SEC, and SAXS.

Native gel electrophoresis under both the ZS and Mg conditions (Figure 5, top panel) indicated that the RVB14 and EVD70 5’CL each exhibited a dominant monomeric structure. However, the PV1 5’CL formed a series of self-associated states, the most dominant of which was apparently larger than a dimer. Over the experimental time frame (~1 week), the self-association state was maintained for each serotype. The self-association of PV1 helps to explain the distinct SEC data for this serotype (Figure 4B), as will be further explored in Section 4.

To detect degradation that could produce additional species in native gels and in SEC, denaturing gels were also run (Figure 5, bottom panel). The RVB14 5’CL showed no discernable degradation. In contrast, after 7 days, the EVD70 5’CL demonstrated a small amount of degradation and the PV1 5’CL showed substantial degradation (~20%), though far less degradation was observed on day one, when the above CD and SEC data were obtained.

### 3.4. SEC-SAXS Identified a Mixture of Monomers and Dimers

Due to the presence of multiple conformational species in solution, the SAXS analysis of individual conformational species required the use of SEC-SAXS. In this approach, an SEC column was used to separate species based on their hydrodynamic radius, and the selected fractions were immediately eluted into a SAXS flow cell. Only relatively intense and well-resolved peaks can be reliably analyzed in this manner. Minimal data analyses were then used to compare the major conformations present for each serotype, and to determine the self-association state.

The beamline restrictions did not allow for the testing of all conditions. In addition, low-/no-salt conditions (i.e., ZS) can be problematic for a SAXS analysis of RNA, due to the resultant intermolecular repulsion that may occur between the negatively charged molecules if the counterion (cation) concentration is insufficient. Therefore, we chose the K+Mg (140 mM KCl + 2 mM MgCl_2_) condition to conduct a parallel SEC-SAXS analysis of all three serotypes. This condition corresponds to the thick dark traces in Figure 3 and Figure 4.

It should be noted, however, that the experimental conditions for SEC-SAXS did differ in some respects from the K+Mg conditions used for CD and SEC. First, the column used for SEC-SAXS was a Superdex 200 (we will refer to these data as SEC200), which should be more adept at the separation of larger species than the Superdex 75 column used to obtain the Figure 5 data (we will refer to these data as SEC75). In addition, the sensitivity limitations of SAXS required a higher concentration (see Table 1), which could have favored dimerization. Also, the time lapse after sample preparation was minimized in the CD and SEC75 data acquisition, in order to minimize sample degradation. In the case of SEC-SAXS, however, approximately 10 days elapsed between the sample preparation and the data analysis at a separate facility. As shown by the gel electrophoresis analysis in Section 3.3, this period of time was sufficient to produce some degradation of the EVD70 sample and significant degradation of the PV1 sample. Due to this degradation, it became challenging to directly compare the peaks between the SEC75 and SEC200 chromatograms, especially for the PV sample.

However, for RVB14, the SEC200 results (Figure 6A) showed a single dominant peak, which is consistent with the analogous SEC75 result (Figure 4A, dark solid line). As shown below (see Figure 7 and Table 3), the SAXS results from this peak indicate a monomer. The position of this monomer peak in Figure 5 (dark solid line) and in Figure 6A helps to align the elution volume scales of the two SEC columns.

The minor shoulder present in Figure 6A near 14 mL indicates a small amount of a larger RVB14 conformational species under K+Mg conditions. A similar shoulder is present near 10 mL in Figure 4A under the same conditions (dark solid line). The fact that this shoulder is somewhat more prominent in Figure 4A than in Figure 6A (which uses a 10× higher concentration) suggests that this larger species is not a dimer. Recall also that gel electrophoresis did not detect significant RVB14 dimerization at the concentration of the Figure 4 SEC data. Therefore, this shoulder, as well as other early-eluting RVB14 peaks (e.g., the ZS peak near 10 mL in Figure 4A), may represent a monomer conformation with a larger hydrodynamic radius than the K+Mg monomer peak.

The dominant SEC200 peak for PV1 (Figure 6B) and EVD70 (Figure 6C) fell in a similar elution position (between 15 and 16 mL) as the single RVB14 peak (Figure 6A). This suggests a similar dominant monomeric conformation for each of the three serotypes under K+Mg conditions. For these three peaks, marked “M” in Figure 6, a comparison of the SAXS profiles (see Figure 7 and Table 3) corroborated this interpretation of similar monomer conformations.

However, multiple SEC peaks were present for PV1 and EVD70 under the K+Mg conditions (Figure 4—dark solid lines—and Figure 6). The peaks at ~16.5 mL in Figure 6B,C are consistent with a hydrolyzed monomer conformation. Hydrolyzed peaks in the Figure 4 data, if present, should appear perhaps near 13 mL. The absence of these peaks corroborates our finding of negligible sample degradation in the Figure 4 data, which was taken on day 1, and negligible degradation of the RVB14 sample in any of these experiments.

The peak marked “D” in Figure 6B was also used for the SEC-SAXS analysis. The SAXS data for this peak (Figure 8 and Table 3) were consistent with a dimer conformation. The three peaks seen between 13 and 14 mL in Figure 6B could then potentially represent a full-length (FL) dimer, a hybrid FL-hydrolyzed dimer, and hydrolyzed PV1 dimers. Similar possible EVD70 dimer peaks are seen in Figure 6C between 13.5 and 14.5 mL. However, the relation between the degradation pattern seen in gel electrophoresis and the SEC chromatograms, as well as the possible contribution of a second monomer conformation with a larger hydrodynamic radius in the PV1 and EVD70 samples, will be further explored in Section 4.

## 4. Discussion

To briefly summarize the results, the ionic environment exerts considerable influence on the conformations of the 5’CL sequences from RVB14, PV1, and EVD70. CD spectroscopy (Figure 3) showed intensity changes and shifts in the maxima and minima. The SEC75 (Superdex 75) chromatograms (Figure 4) showed a general decrease in the hydrodynamic radius with the addition of salt, though this pattern differed slightly for RVB14 vs. EVD70, and the PV1 5’CL displayed a quite different pattern. These serotype differences were at least in part due to differences in the dimerization propensity: PV1 strongly dimerizes, while EVD70 and especially RVB14 show little propensity to form dimers (Figure 6 TOP panel). We found a higher degradation propensity for PV1 and EVD70 than for the RVB14 5’CL (Figure 5 bottom panel). SEC-SAXS using a Superdex 200 column (SEC200 data; Figure 6) showed the effects of dimerization and degradation after 10 days under the conditions of 140 mM NaCl and 2 mM MgCl_2_. The SAXS analysis of four peaks from these SEC200 data indicated a similar dominant monomer conformation for each of the three serotypes, and a distinct dimer for a peak drawn from an early eluting PV1 peak. Notably, the simultaneous presence of multiple conformational species was undeniable. Some, but not all, of this multiplicity was due to dimerization. We explore these observations in more detail below.

### 4.1. Sequential Differences Among the Three 5’CL Sequences 

The yellow shaded areas in Figure 2 highlight the main sequential and secondary structure differences among the 5’CL sequences from RVB14, PV1, and EVD70. The stars in Figure 2 denote self-complementary sequences within loop regions that may contribute to dimerization via kissing complex formation. PV1 (Figure 2B) is unique among these three serotypes in containing a self-complementary stretch of four nucleotides in the loop of SLD. This is almost certainly the origin of the strong dimerization of the PV1 5’CL. RVB14 and EVD70 contain no self-complementary loop sequences longer than two nucleotides. The slightly higher dimerization propensity of EVD70 vs. RVB14 may be related to the fact that the EVD70 self-complementary sequences are each CG, while the RVB14 sequences are each UA or AU. GC base pairs are, of course, more stable than AU base pairs. Note that the four-nucleotide self-complementary stretch in the PV1 SLD loop is also GC-rich (CGCG), further strengthening its ability to drive dimerization. The additional unpaired GC sequence in the PV1 SLC loop may be responsible for the higher-order structures observed in native gel electrophoresis (Figure 5, top panel), which were filtered out prior to CD, SEC, and SAXS.

Explaining the differences in the degradation propensity among the serotypes is more difficult. First, although we reproducibly observed the least degradation propensity in the RVB14 5’CL samples, variability was observed from sample to sample, and so an element of chance cannot be excluded. However, if the PV1 and EVD70 are indeed more susceptible to degradation, this may be related to the larger junction region (see the additional GC sequence highlighted at the junction of Figure 2B,C), which may produce a hydrolysis-susceptible unpaired loop. The PV1 and EVD70 5’CL sequences both also have two additional base pairs in SLB relative to RVB14, and PV1 and EVD70 both have a tetraloop in SLD vs. the triloop of RVB14. Interestingly, a positive correlation existed among the following sequence factors: the SLB length, the junction size, and the presence of an SLD tetraloop (see [23] and data therein). Further experiments would be needed to determine with certainty the link between the sequence and chemical stability. It should be noted that partially degraded or otherwise shortened 5’CL sequences have been detected in vivo, and that some of these shortened sequences are thought to be active in the initiation of replication [24,25,26,27,28]. For example, it was found that coxsackievirus B3 (CVB3) will primarily utilize strands of the viral genome with 5’ terminal deletions throughout a persistent infection to avoid triggering a host immune response [27]. Therefore, degradation patterns may be more than simply an experimental artifact. Much the same could be said for 5’CL dimerization.

### 4.2. Hydrodynamic Radius

Some of the observed differences in the hydrodynamic size, made evident by peak positions in SEC chromatograms, were clearly due to dimerization. Most notably, the anomalous position of the dominant PV1 peak under K+Mg conditions (dark solid trace in Figure 4B) was almost certainly due to dimerization, while the smaller peak near 12 mL appeared to be a monomer with similar characteristics as the dominant RVB14 and EVD70 peaks present under these conditions (dark solid traces in Figure 4A,C, respectively). This PV1 dimer peak appeared to have been split into three peaks in the SEC200 data (Figure 6B), perhaps due to the sample degradation that took place between the time of the two SEC data sets. It would appear that most PV1 degradation fragments shared a similar hydrodynamic radius, since their monomers were within one (albeit broad) peak near 16.5 mL in Figure 6B. Thus, the dimers that formed would have lain in three different hydrodynamic size ranges: FL–FL, FL–fragment, and fragment–fragment.

The EVD70 5’CL also showed signs of degradation after 7 days (see the monomer peak near 16.5 mL in Figure 6C), but degradation and dimerization were both less prevalent than for PV1 (see Figure 5). Due to a combination of these factors, considerably less relative intensity of dimers was observed in Figure 6C vs. Figure 6B. Very little dimer intensity was observed for RVB14 (see small shoulder near 14 mL in Figure 6A).

However, shifts to a lower volume in the SEC data could also be due to the presence of a less compact monomer structure. This *was* the case with RVB14: we observed an insufficient dimerization propensity for the RVB14 5’CL to account for the peak near 10 mL in the ZS condition (solid light trace in Figure 4A). Moreover, a previously published SAXS-based analysis of RVB14 under similar conditions clearly indicated an “open” monomer conformation, with a relatively large hydrodynamic radius ([29], see next section).

The most likely explanation for the RVB14 peaks under the remaining intermediate conditions in Figure 4A is then relatively straightforward. The two peaks observed under K conditions (short dashes) represent a compact and non-compact monomer, with the breadth and partial overlap of these two peaks suggesting an interconversion rate that is slightly slower than the SEC time frame (minutes). The non-compact monomer (near 10 mL) appeared to be dominant under K conditions. Under Mg conditions (long dashes), a single peak was observed, intermediate between the two peaks just discussed: the interconversion rate between the two monomeric conformations apparently increases in the presence of magnesium, producing a single time-averaged peak. The position of this peak, relative to the weighted average of the two peaks under K conditions, suggests that magnesium also shifts the equilibrium towards the compact structure (to the right). The low shoulder near 10.5 mL under K+Mg conditions (dark solid line in Figure 4A) was therefore likely due to the presence of a small amount of non-compact monomers in equilibrium with the dominant compact monomer near 11.5 mL.

A similar analysis of the EVD70 SEC75 data (Figure 4C) yielded similar conclusions, although each of the conditions showed a clear preference for monomer type: K and K+Mg both favored the compact monomer, while Mg and ZS favored the non-compact monomer. That is, potassium is the deciding factor for the equilibrium distribution of the EVD70 5’CL. However, an additional shoulder was seen near 9 mL. Since this peak elutes earlier than the non-compact monomer, this may be due to a small amount of dimers or even higher-order self-association that survives or forms after filtration. A similar, but more intense, shoulder in the PV1 data would then be consistent with the higher-order PV1 structures observed via gel electrophoresis (Figure 5).

Some questions remain. Is it coincidence that the PV1 dimer peak elutes at a similar volume as the non-compact monomer RVB14 and EVD70 peaks (near 10–10.5 mL in Figure 4)? The separation of non-compact monomers from dimers in the PV1 SEC75 chromatograms may then be difficult or impossible: the dominant PV1 peak under K+Mg conditions in Figure 4B, for instance, may be a mixture of both species. It is possible that SEC200 is better able to separate these species, and this could be part of the reason for the large number of peaks in Figure 6B (in addition to degradation over time). SEC200 also may be better able to separate different dimer species. Dimers consisting of two compact conformations, two non-compact conformations, or one of each could be envisioned. Is this the origin of the three PV1 “dimer” peaks in Figure 6B? So, in brief, not all questions have been answered at this point. In future analyses, techniques such as SHAPE [30] and nuclease footprinting [31] may be utilized to answer some of these questions. For example, the SHAPE technique can identify unpaired bases [30,32]. The additional use of higher-resolution approaches such as X-ray crystallography and the NMR/SAXS approach (see next section) could also provide further refinement and a deeper understanding of 5’CL conformations and interconversions.

However, the present analysis is sufficient to establish that at least two monomeric conformations and one or more dimer conformations can be present in 5’CL RNA samples, with equilibrium distributions and interconversion rates that depend upon both the ionic conditions and the serotype (sequence).

### 4.3. Previously Reported 5’CL Structural Analysis

A previous examination of the tertiary structure of the enterovirus 5’CL produced three potential monomer conformations. In a 2019 study, the solution structure of the rhinovirus B14 (RVB14) 5’CL was examined using a combined NMR and small-angle X-ray scattering (SAXS) approach. Both the ZS and Mg conditions were explored. Consistent with the above analysis, dimerization was not detected under either condition. Under the ZS condition, an “open” conformation was reported, with the two longest stem loops (SLB and SLD) positioned approximately perpendicular to each other (Figure 9A). In contrast, under the Mg condition, a “closed” or compact structure was reported, with SLB and SLD parallel and side by side.

The “open” structure of Figure 9A is consistent with the “non-compact” RVB14 monomer discussed above (solid light trace in Figure 4A). The “closed” structure of Figure 4B is consistent with the “compact” RVB14 monomer discussed above (e.g., the solid dark trace in Figure 4A). However, recall that, under Mg conditions (long dashed trace in Figure 4A), the most likely explanation for the peak position is rapid equilibrium between the compact and non-compact structure, with the compact structure being dominant. Apparently, this dominance was sufficient to produce a closed structural model in the previous NMR/SAXS analysis under the Mg condition.

In recent crystallographic studies from two independent teams, the conformation of the 5’CL from four enterovirus serotypes was determined to a high resolution. In 2023, Das et al. reported the structure determination of the 5’CL (Figure 9C) from coxsackievirus B3 (CVB3) [33]. Also in 2023, Gottipati et al. reported the structure determination of the 5’CL from CVB3 and PV1 (Figure 9D) [34]. Das et al. later added an analysis of the RVB14 and RVC15 5’CL structures [35]. Each study found a “closed” structure, with SLB oriented anti-parallel to SLD. Note that crystallization required either the mutation of loops from SLB or SLD to facilitate co-crystallization with an antibody fragment (Fab) [33,35] or the fusion of a tRNA molecule to the 5’CL termini [34].

The “closed” crystal structures are consistent with the “compact” monomer conformation discussed above (e.g., the Figure 6 monomer peaks). Neither crystallography group reported a second structure, such as the “open” structure found in solution. This is not surprising, since the presence of salts was required for crystallization. In addition, an “open” structure is more irregular and possibly also more dynamic, and hence would be more difficult to crystallize. The “closed” structures modeled from both the SAXS/NMR and crystallography data did, however, differ in the arrangement of helices—most notably, the parallel vs. anti-parallel alignment of SLB and SLD, as discussed above. This difference can also essentially be expressed as a different packing arrangement; the crystal models take on an H arrangement (see Figure 9E), while the “closed” SAXS/NMR model contains a cH arrangement (see Figure 9F).

### 4.4. The Crystal vs. Solution Approaches

The ability of ionic conditions to affect an RNA tertiary structure is well established [36,37,38]. The primary reason is that the RNA phosphodiester backbone carries multiple negative charges, which cause electrostatic repulsion between stem loops. This typically prevents the close approach of stem loops in the tertiary structure. This repulsion can, however, be neutralized by the presence of monovalent (e.g., K^+^) or divalent (e.g., Mg^++^) cations. For this reason, an RNA structure is often determined under a variety of ionic conditions to detect alternative tertiary arrangements.

These influences are plain to see in both the previously reported structures and in the new data presented here. What also becomes clear is that the choice of technical approach can influence the results. The NMR/SAXS approach as used above provides an excellent relative alignment of the helical axes in solution via the use of residual dipolar couplings (RDCs) together with SAXS profiles. However, the RDC-based approach contains ambiguity regarding helical orientation; that is, parallel vs. anti-parallel helical arrangements are difficult to distinguish. The “open” and “closed” structures above therefore accurately describe the overall shape of the two conformations (compact and non-compact) that are present in solution. Additional uncertainty can be introduced by the fact that multiple conformations may be simultaneously present. SEC-SAXS can help to separate these conformations, but as we have seen, the complete separation of conformations is not always possible.

The crystallographic approach inherently provides a higher resolution, and is able to determine not only the helical alignment, but also the helix orientation. This approach can unequivocally state that the crystallized 5’CL conformation aligns SLB and SLD in an anti-parallel orientation (an H model). It should be noted, though, that the crystal approach only explored high-ionic-strength environments and required mutation, co-crystallization with a non-natural partner, or fusion with a tRNA, together with precipitating agents and crystal contacts, all of which can affect the conformation. Crystallography also presents the double-edged sword of selective crystallization: a mixture of conformations may lead to the growth of a crystal containing only one conformation. This is an advantage: it serves to isolate a single conformation for a high-resolution structural study. However, this can also lead to the masking of additional conformations that may be present in solution.

## 5. Conclusions

This work presents a biophysical analysis of the conformation of the 5’CL sequences from three enterovirus serotypes under four distinct ionic conditions. In general, the hydrodynamic radii of the 5’CL monomers decreased with an increase in salts. The findings are consistent with the presence of an open and a closed monomeric conformation, in an equilibrium that is influenced by the ionic conditions. The ionic environment also affects self-association, with salts particularly driving the PV1 5’CL to form dimers and higher-order structures. These results indicate that a combined crystal and solution approach is necessary for the development of a more complete picture of the 5’CL structure under native conditions that may aid in the development of effective anti-viral therapeutics.

## Figures and Tables

**Figure 1 viruses-17-01127-f001:**
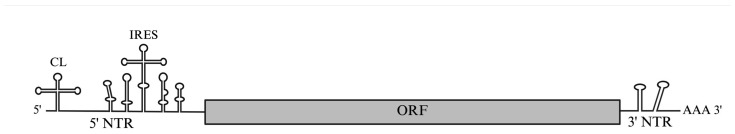
Schematic diagram of the enterovirus genome. The enterovirus genome consists of a single-stranded RNA molecule with three major regions: one open reading frame (ORF) flanked by two non-translated regions (NTRs). The 5’NTR includes the cloverleaf (CL) and the internal ribosome entry site (IRES), which are utilized in replication and translation, respectively.

**Figure 2 viruses-17-01127-f002:**
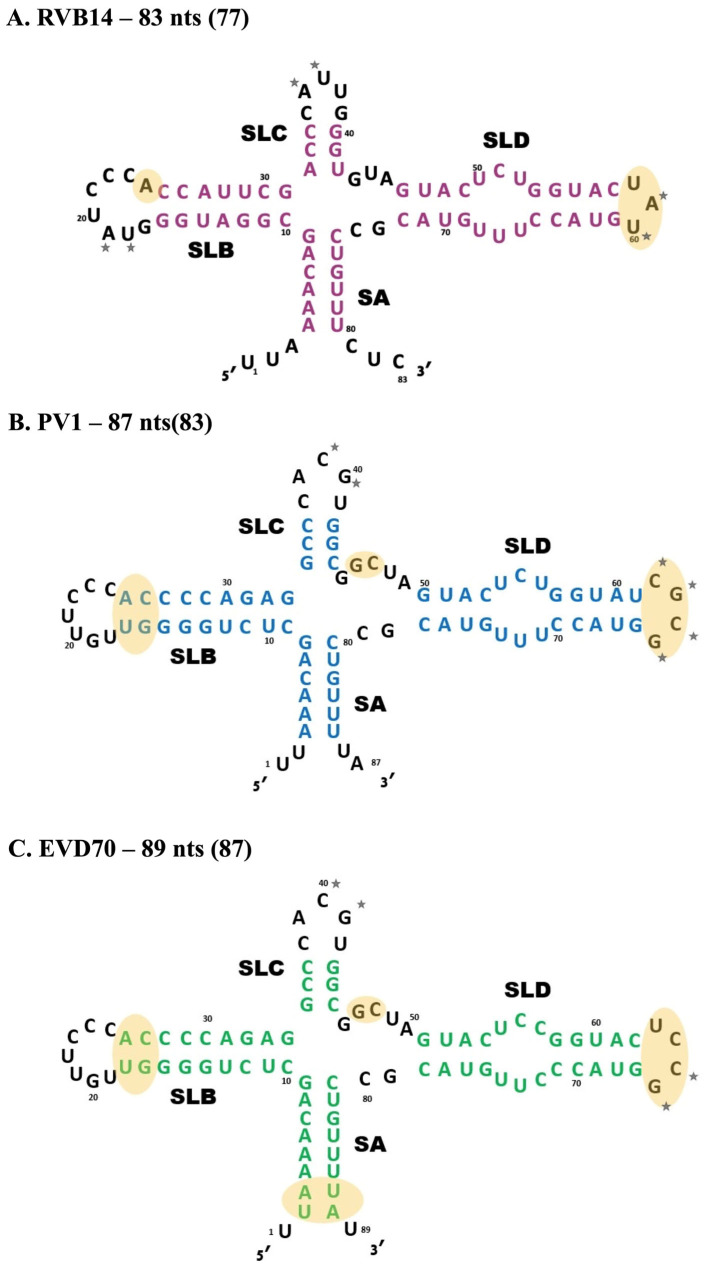
Secondary structure models of the three 5’CL sequences used in this study. The enterovirus serotypes utilized in this manuscript were (**A**) RVB14, (**B**) PV1, and (**C**) EVD70. The 5’CL coloring seen here (pink, blue, and green, respectively) will be propagated through the remaining figures. Stem A (SA) and three stem loops (SLB, SLC, and SLD) are labeled. Self-complementary motifs in loop regions with the potential to drive dimerization are marked with stars, while key differences among the three 5’CL sequences are highlighted in yellow. The number of nucleotides in each molecule is shown, along with the number of nucleotides excluding the unpaired terminal bases (in parentheses).

**Figure 3 viruses-17-01127-f003:**
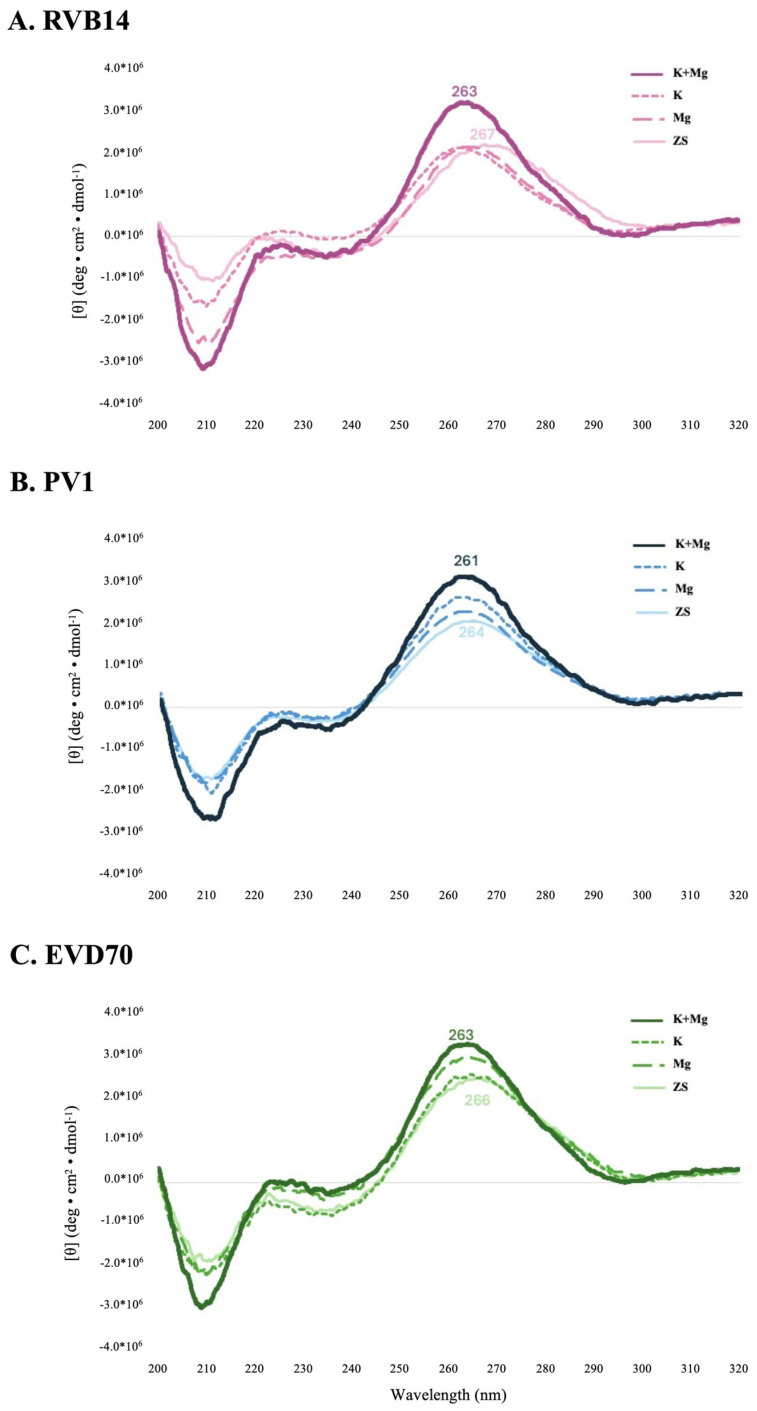
CD spectra of the 5’CL vs. the ionic environment. The 5’CL from each serotype was investigated under the four ionic conditions of Table 1 (ZS, Mg, K, and K+Mg) to detect conformational changes. The serotype and ionic conditions are labeled in the figure. The wavelength (in nm) of maximum ellipticity is marked on the individual traces for the zero-salt (ZS) and the K+Mg conditions. The RNA concentration was 2 × 10^-3^ mM.

**Figure 4 viruses-17-01127-f004:**
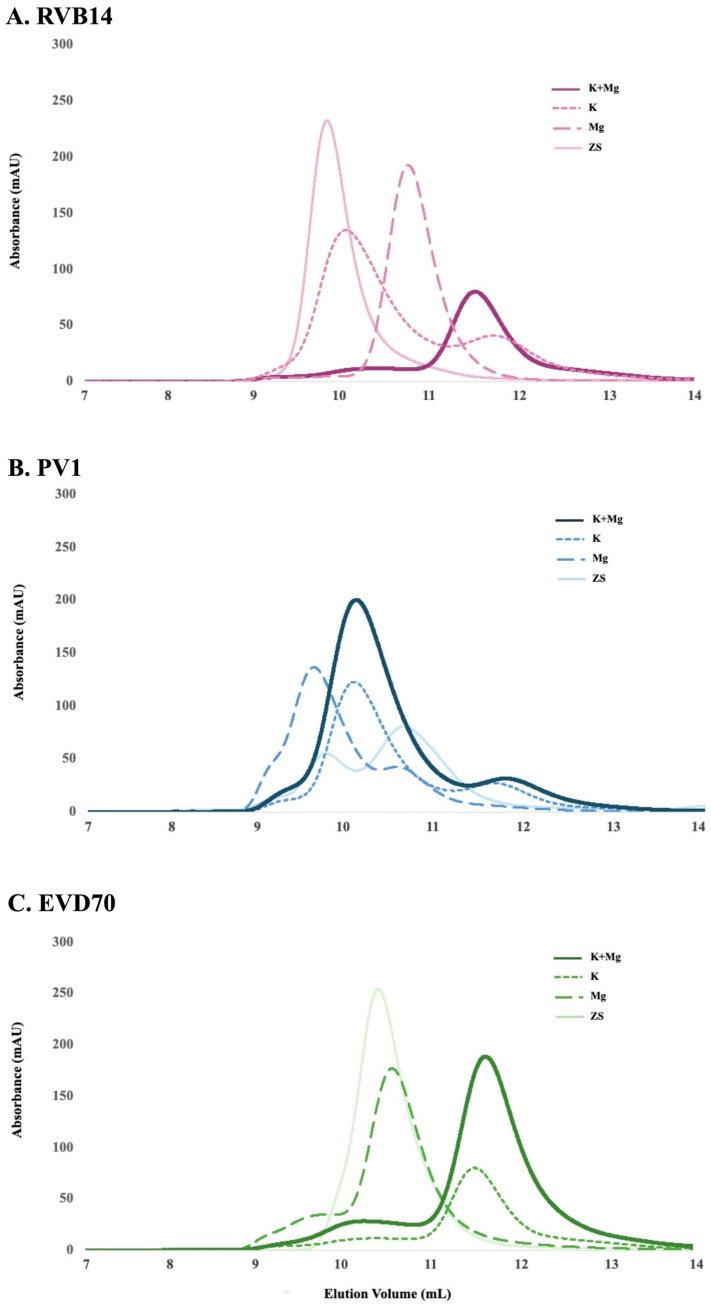
SEC75 chromatograms of the 5’CL vs. the ionic environment. The 5’CL from each serotype was investigated under the four ionic conditions of Table 1 using a Superdex 75 Increase 10/300 column.

**Figure 5 viruses-17-01127-f005:**
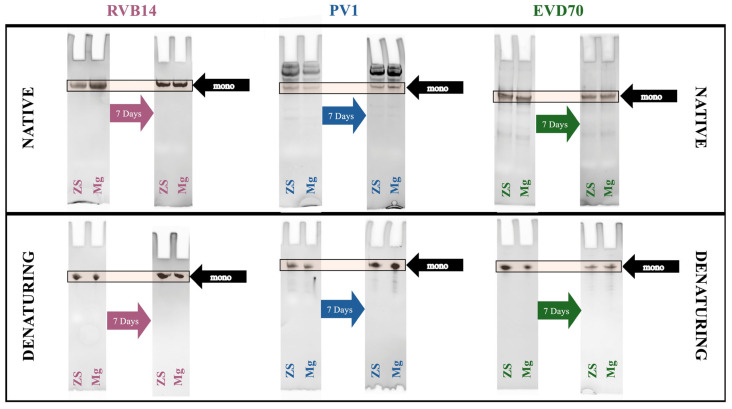
Native and denaturing gel electrophoresis of 5’CL. The 5’CL from each serotype was investigated using native (10% polyacrylamide; **top panel)** and denaturing (15% urea-polyacrylamide; **bottom panel**) gel electrophoresis. Experiments were performed on day 1 after sample preparation, for comparison with SEC75 data (see Figure 4). Samples were then stored at 4 °C for 7 days and gel electrophoresis was again performed, for comparison with SEC200 data (see Figure 6). Serotype, 7-day storage, and ionic conditions (ZS or Mg) are labeled in figure. Black arrows indicate position of 5’CL monomer. RNA concentration was 0.01.

**Figure 6 viruses-17-01127-f006:**
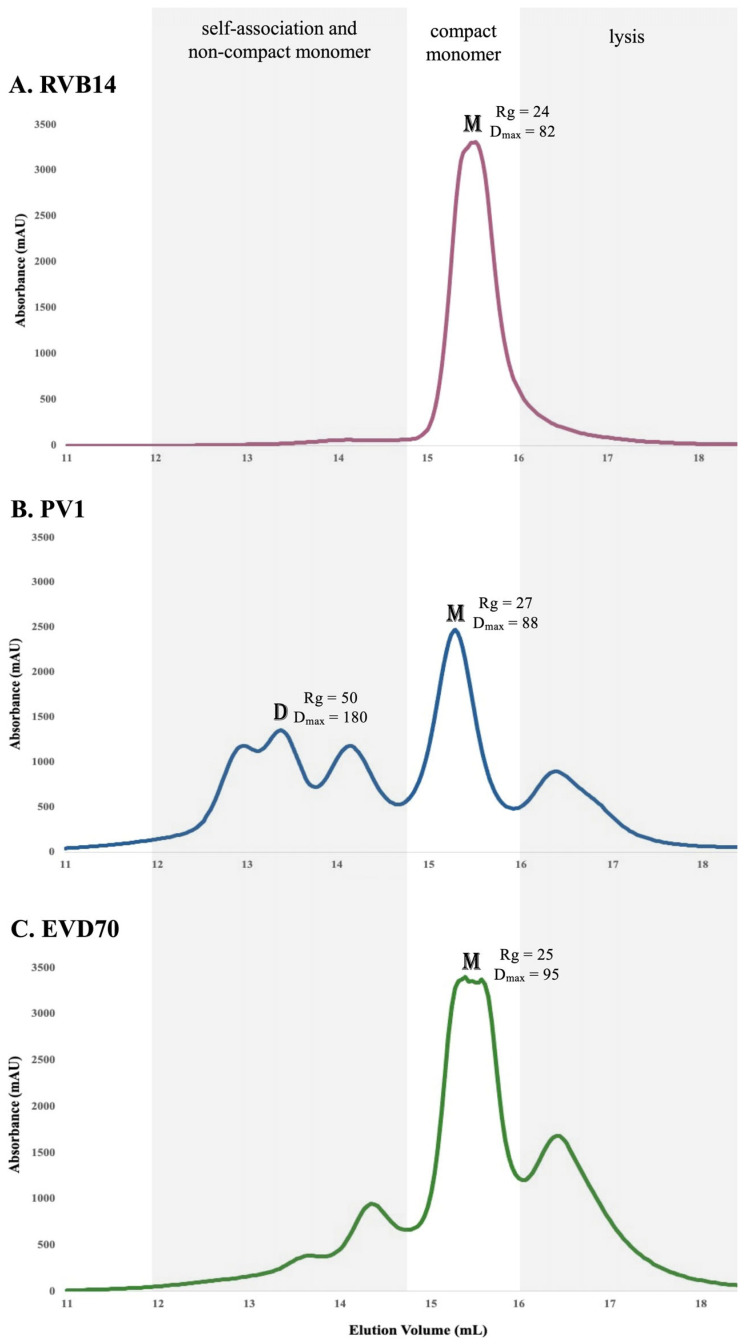
SEC200 chromatograms of 5’CL under K+Mg conditions. The 5’CL from the three serotypes (**A**) RVB14, (**B**) PV1 and (**C**) EVD70 were investigated under the conditions of 140 mM KCl and 2 mM MgCl_2_, utilizing a Superdex 200 Increase 10/300 column. The peaks marked “M” and “D” were selected for the SAXS analysis (see Figure 7 and Figure 8). The parameters from the SAXS analysis are printed next to each analyzed peak. Shading is used to roughly delineate the regions of the chromatograms that correspond to self-associated states or non-compact monomers, compact monomers, or lysis products. The RNA concentration was 0.2 mM.

**Figure 7 viruses-17-01127-f007:**
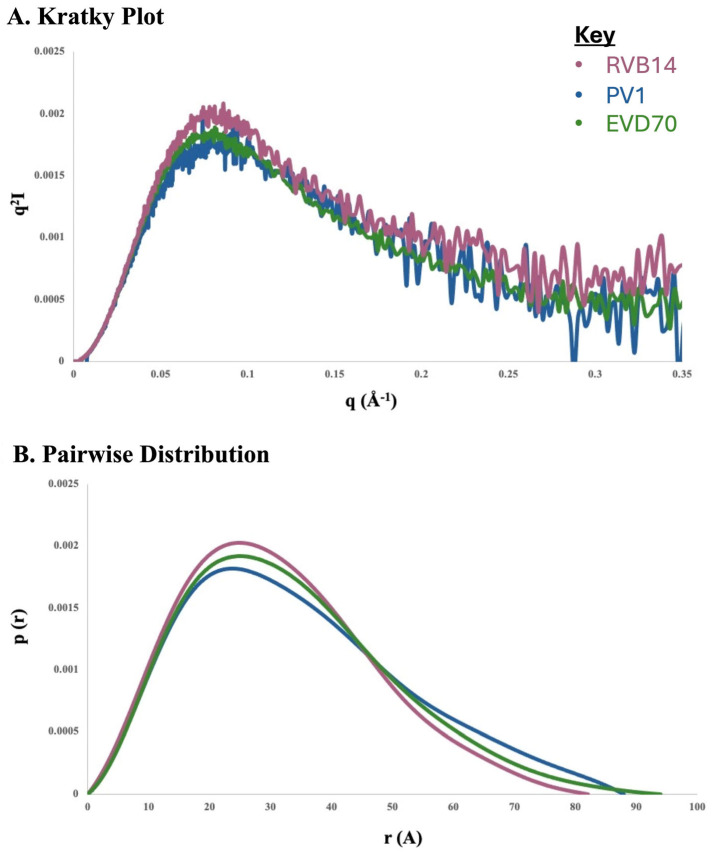
SAXS data of the selected monomeric SEC200 peaks. The three peaks marked by “M” in Figure 6 were injected directly into the SAXS flow cell to produce the scattering profiles above. (**A**) Kratky plots; (**B**) pair distance distribution function plots. The bell shape of the Kratky plots in Figure 7A indicates that these RNA molecules were well folded under K+Mg. The data from the three serotypes are overlayed and color-coded as defined within the figure.

**Figure 8 viruses-17-01127-f008:**
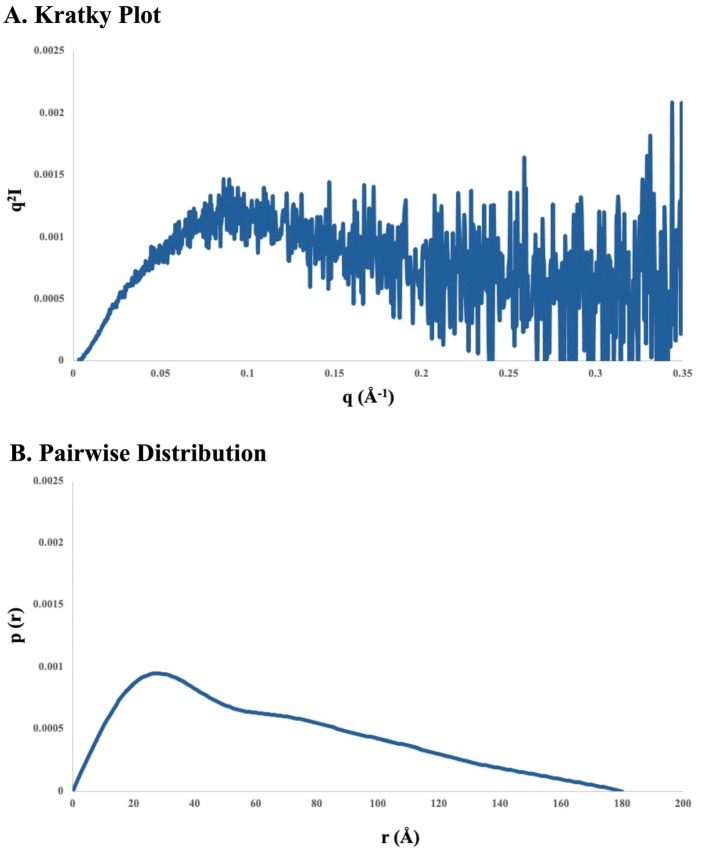
SAXS data of the selected dimeric PV1 SEC200 peak. The PV1 peak marked by “D” in Figure 6 was injected directly into the SAXS flow cell to produce the scattering profiles above. (**A**) Kratky plot; (**B**) pair distance distribution function plot. Note the extension of the pairwise distribution to ~180 Å vs. ~90 Å in the data from Figure 6.

**Figure 9 viruses-17-01127-f009:**
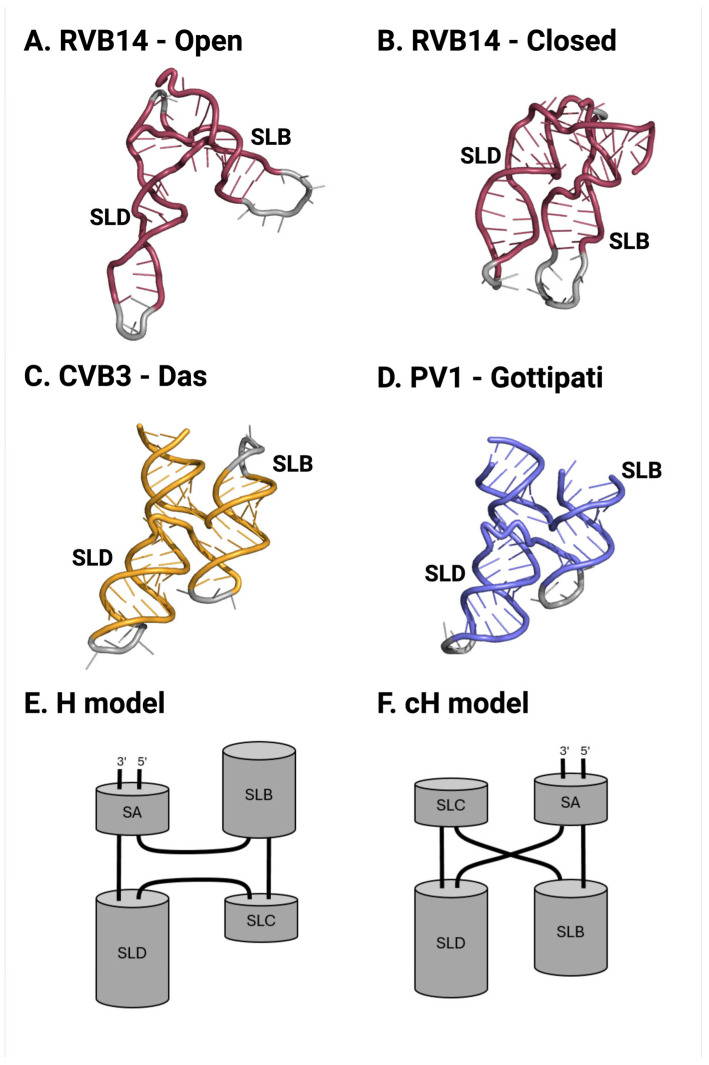
Structural models of select Enterovirus 5’CL. The RVB14 5’CL was studied via solution techniques [29]. Two distinct conformations were reported: (**A**) an “open” non-compact conformation under ZS conditions and (**B**) a “closed” compact conformation under Mg conditions. Crystallography was used to determine the structural models for (**C**) the CVB3 5’CL [33] and (**D**) the PV1 5’CL [34], as well as the other serotypes discussed in the text, in a strongly ionic environment. Each of the crystal structures was a “closed” conformation. (**E**) A schematic “H” helix packing model consistent with the crystal structures. (**F**) A schematic “cH” helix packing model consistent with the solution structure model of part B.

**Table 1 viruses-17-01127-t001:** Ionic conditions used for analysis.

	Potassium Chloride (K+)	Magnesium Chloride (Mg2+)
ZS	0 mM	0 mM
Mg	0 mM	5 mM (CD = 2 mM)
K	140 mM	0 mM
K+Mg	140 mM	2 mM

All samples contain 10 mM NaPi and 0.1 mM EDTA.

**Table 2 viruses-17-01127-t002:** RNA Concentrations used for each Analysis Method.

	CD	Gels	SEC75	SEC200	SAXS
Concentration	0.0015 mM	0.01 mM	0.02 mM	0.2 mM	0.2 mM

**Table 3 viruses-17-01127-t003:** SAXS data analysis.

Serotype	RVB14	PV1		EVD70
Conditions	Monomer	Monomer	Dimer	Monomer
R_g_ (Å)	24.1463	26.8774	49.9719	25.3639
I(0) (cm^−1^)	0.0356	0.0212	0.0257	0.0661
D_max_ (Å)	82	88	180	95
χ^2^	0.925	1.008	1.036	1.108

## Data Availability

The data for this manuscript will be made available upon request.

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
