# Peer review of "The Importance of Solution Studies for the Structural Characterization of the Enterovirus 5’ Cloverleaf"

_viruses, 2025, doi:10.3390/v17081127_

Round 1

Reviewer 1 Report

Comments and Suggestions for Authors

The manuscript is written well, but the experimental results and the conclusions contradict in some parts. I would suggest acceptance of this manuscript after addressing these queries properly:

  Results:  a. Why were K and Mg chosen for comparison? b. 'Both RVB14 and EVD70 (Fig. 3C) appears to be more strongly affected by Mg than by K'. Can you explain it? c.  "multiple SEC peaks" is stated, which contradicts "two monomeric" and "dimeric" conformations, as you mentioned earlier. Can you elaborate on it? d. From native gels, were any of these higher-order species present in the SEC/CD/SAXS samples, or were they completely filtered out for biophysical experiments? e. What is larger than a dimer (from native gel) for PV1? f. Can the mol. wt. marker be added in Figure 5 for better visualisation? g. Were minor populations observed in SEC but not fully characterized by SAXS? h. Discussion: The first two paragraphs read and mean almost the same. Can you rectify/remove one?  i. Did you deposit the SAXS data in SASBDB database? It's not mandatory, just a suggestion.   Abstract: I. Minor suggestion: Could you specify the biophysical techniques utilised in the abstract, i.e. SAXS, CD, SEC? II. The authors claim that 'The 5’CL conformation may be more complex than the current literature suggests'. Why is it so?

Author Response

Comment #1: Why were K and Mg chosen for comparison?

Response #1: K and Mg are present in cells at concentration ranges similar to those used in this study. The free intracellular concentration of other cations is generally lower and may not supply enough charge to effectively neutralize the backbone charge. In addition, K or Mg concentrations utilized were chosen to match experimental conditions for some of the previously published models, facilitating comparison.

Comment #2: 'Both RVB14 and EVD70 (Fig. 3C) appears to be more strongly affected by Mg than by K'. Can you explain it?

Response #2: This fact is clear from both CD (Fig 3) and SEC (Fig 4). This is likely due to a divalent cation more easily concentrating the positive charge necessary to neutralize the multiple negative charges of the RNA backbone. This is addressed in the discussion (lines 178 - 184) 

Comment #3: "multiple SEC peaks" is stated, which contradicts "two monomeric" and "dimeric" conformations, as you mentioned earlier. Can you elaborate on it?

Response #3: By “multiple SEC peaks”, we mean more than one. At line 193, we state multiple peaks in Figure 4 “under most conditions”, which is certainly true. At line 296, we state multiple peaks “for PV1 and EVD70 under K+Mg conditions” in Figure 6, which is also true. In line 431, we state two monomeric and one or more dimer conformations in general, considering all three serotypes and four conditions, “can be present”. This last is a summary statement, and not a specific statement about a specific serotype under a specific ionic condition. For instance, there is little evidence of significant multiple conformations of RVB14 under K+Mg condition, but RVB14 takes on multiple conformations under some conditions (see Fig 3A).

Comment #4: From native gels, were any of these higher-order species present in the SEC/CD/SAXS samples, or were they completely filtered out for biophysical experiments?

Response #4: 220 um filters were used to prepare samples for SEC/CD/SAXS. This is described in the Materials and Methods. This filtering removes the larger species. We’ve inserted a mention of this fact in lines 116 to 117.

Comment #5: What is larger than a dimer (from native gel) for PV1?

Response #5: This higher order association could be non-specific aggregation, however, we don’t have specific data on this at this time.

Comment #6: Can the mol. wt. marker be added in Figure 5 for better visualisation?

Response #6: For the size of RNA we are using, we could not find a commercial RNA ladder. Instead, we use old samples from our laboratory as standards, as well as experience of where bands tend to fall. These internal laboratory standards would not be informative to include, but we are certain of where the monomer band lies, via years of experience. Presumably the next highest band is dimer (since SEC-SAXS does indicate the presence of dimer), but then additional bands must simply be described as higher order.

Comment #7: Were minor populations observed in SEC but not fully characterized by SAXS?

Response #7: Correct, SEC-SAXS requires at least mostly non-overlapping of peaks  and significant amount of material (i.e. very minor and overlapping peaks are difficult ot analyze). We also faced limitations due to available synchrotron time, but within these limitations, we chose to analyze key peaks that we feel establish the pattern of structures described in the manuscript.

Comment #8: Discussion: The first two paragraphs read and mean almost the same. Can you rectify/remove one? 

Response #8: The intention of the first paragraph in the DISCUSSION section is to briefly summarize the results, that would then be dissected in the following sub-sections. To emphasize this, we have added “Briefly Summarizing the Results:” to the top of the first paragraph of the DISCUSSION, so that the reader understands our approach. To further clarify, the last sentence of the paragraph states that “We now explore these observations in more detail. “

Comment #9: Did you deposit the SAXS data in SASBDB database? It's not mandatory, just a suggestion.  

Response #9: Thank you, we will look into that database.

Comment #10: Minor suggestion: Could you specify the biophysical techniques utilised in the abstract, i.e. SAXS, CD, SEC?

Response #10: Please see the change to the abstract (Line 14-15).

Comment #11: The authors claim that 'The 5’CL conformation may be more complex than the current literature suggests'. Why is it so?

Response #11: Current literature suggests three potential monomer models, under essentially the conditions that we call ZS, Mg or K, with one model per condition. We have shown that multiple conformations can exist simultaneously in solution, including a dimer, and have presented evidence in support of interconversion between conformations.

Reviewer 2 Report

Comments and Suggestions for Authors

As for the content of the manuscript, there are several parts that are not entirely clear. Unfortunately, according to the description of the experiments, I could not unambiguously repeat all the measurements.

For example, I do not understand this sentence (L 94-95), although I know what the authors meant to say; "All pure samples were combined and ethanol precipitated with 0.1 volumes of 3M sodium acetate and 2-3 volumes of 190% ethanol overnight at -20 °C."

Although the authors compare their results with those of other analyses, their conclusions would be strongly supported by the results obtained, for example, by nuclease footprint analysis.

What I would certainly recommend, however, is to include CD differential spectra in the manuscript, where the influence of ions on CL conformation would be much more obvious.

Author Response

Comment #1: As for the content of the manuscript, there are several parts that are not entirely clear. Unfortunately, according to the description of the experiments, I could not unambiguously repeat all the measurements. For example, I do not understand this sentence (L 94-95), although I know what the authors meant to say; "All pure samples were combined and ethanol precipitated with 0.1 volumes of 3M sodium acetate and 2-3 volumes of 190% ethanol overnight at -20 °C."

Response #1: Experimental procedures were updated for clarity and to match related work. 

Comment #2: Although the authors compare their results with those of other analyses, their conclusions would be strongly supported by the results obtained, for example, by nuclease footprint analysis.

Response #2: This technique may prove to be advantageous for confirmation and in addressing further structural questions that arose from this project. Please see lines 412-414 where we mention the potential application of this and other techniques in future studies.

Comment #3: What I would certainly recommend, however, is to include CD differential spectra in the manuscript, where the influence of ions on CL conformation would be much more obvious.

Response #3: We have added CD differential analysis as Supplemental Figure S1.

Reviewer 3 Report

Comments and Suggestions for Authors The work titled "The Importance of Solution Studies for Structural Characterization of the Enterovirus 5’ Cloverleaf" by Morgan G. Daniels et al. presents a biophysical investigation of the 5′ Cloverleaf (5′CL) RNA structure in three enteroviruses under various ionic environments, utilizing techniques including CD, SEC, and SAXS. The study provides novel insights into conformational heterogeneity and dimerization propensities among serotypes, specifically PV1, RVB14, and EVD70. This work fills a knowledge gap in RNA structural dynamics relevant to viral replication and could inform future therapeutic targeting strategies. The experimental design is robust, and the data interpretation is well-considered. The study falls well within the scope of a virology-focused journal, offering valuable insights into RNA structural dynamics relevant to viral replication. I recommend this manuscript for publication, pending minor revisions to address the concerns outlined below.   1. To clarify the conformational states and dynamics of the 5′CL RNA, the authors could consider performing SHAPE probing. This would enable detailed mapping of nucleotide flexibility for direct comparison between monomeric and dimeric forms, as well as between compact and non-compact conformations.    2. The Discussion section has repeated paragraphs (lines 312–325 are duplicated from 287–301). Please remove redundancy.

Author Response

Comment #1: To clarify the conformational states and dynamics of the 5′CL RNA, the authors could consider performing SHAPE probing. This would enable detailed mapping of nucleotide flexibility for direct comparison between monomeric and dimeric forms, as well as between compact and non-compact conformations.   

Response #1: This technique may prove to be advantageous for confirmation and in addressing further structural questions that arose from this project. Please see lines 434-440 where we mention the potential application of this and other techniques in future studies.

Comment #2: The Discussion section has repeated paragraphs (lines 312–325 are duplicated from 287–301). Please remove redundancy.

Response #2: The first paragraph of the DISCUSSION is a brief summary of the results. We felt it was helpful to gather the main points here, and then to dissect each point in the following sub-sections. To emphasize this, we have added “Briefly summarizing the results:” to the top of the first paragraph of the DISCUSSION, so that the reader knows our approach. To further clarify, the last sentence of the paragraph states that “We now explore these observations in more detail. 

Round 2

Reviewer 2 Report

Comments and Suggestions for Authors

The authors made some technical changes, but essentially did not improve their manuscript. Although the results and overall idea are interesting, in my opinion, their presentation could be modified in such a way as to address a broader scientific community. Personally, I expected a more significant change.

Nevertheless, the authors have made improvements based on my comments, and I therefore agree that the article should be accepted.